# New Therapeutic Targets TIGIT, LAG-3 and TIM-3 in the Treatment of Advanced, Non-Small-Cell Lung Cancer

**DOI:** 10.3390/ijms26094096

**Published:** 2025-04-25

**Authors:** Jacek Kabut, Anita Gorzelak-Magiera, Iwona Gisterek-Grocholska

**Affiliations:** Department of Oncology and Radiotherapy, Medical University of Silesia, 40-514 Katowice, Poland; anitagor@op.pl

**Keywords:** lung cancer, TIM-3, LAG-3, TIGIT, immunotherapy

## Abstract

The introduction of immunotherapy and target therapy into clinical practice has become a chance for many patients with cancer to prolong their survival while maintaining optimal quality of life. Treatment of lung cancer is excellent evidence of the progress of medical therapies. An understanding of the mechanisms of tumor development has led to the evolution of new methods of treatment. Immunoreceptors of T cells with the immunoglobulin domain ITIM, TIM-3 (T-cell immunoglobulin- and mucin domain-3-containing molecule 3), and LAG-3 (lymphocyte activation gene-3) represent new interesting therapeutic targets. The combination of anti-PD-1 and anti-CTLA-4 blockade has proven the possibility of strengthening the anti-tumor response by acting via two separate mechanisms. Adding additional checkpoints to the PD-1 blockade offers hope for further improvements in the effects of the treatment of patients and expanding the group responding to immunotherapy. This paper presents new promising molecular targets along with studies demonstrating the treatment results using them.

## 1. Introduction

Lung cancer is one of the biggest health challenges worldwide. In 2022, approximately 2.48 million new cases were diagnosed and 1.8 million deaths, respectively [1]. Non-small-cell lung cancer (NSCLC) is the major histological lung cancer subtype, accounting for approximately 85% of all lung cancer cases [2]. Patients diagnosed with localized lung cancer have a better prognosis, but the percentage of early diagnoses is low. The majority of patients with NSCLC are diagnosed in stage III or IV of the disease [3]. The survival in this group of patients is unsatisfactory. Before the era of modern therapies, the use of chemotherapy containing platinum allowed for achieving median survival of 7.9 months [4]. The introduction of immunotherapy into clinical practice has become a chance for many patients with cancer to prolong their survival while maintaining optimal quality of life. The revolution in clinical practice was initiated by the registration by the FDA in 2011 of the first checkpoint inhibitor—ipilimumab. Since then, we have witnessed the dynamic development of immunotherapy in subsequent clinical indications. Anti-CTLA-4 (cytotoxic T-lymphocyte antigen-4), anti-PD-1(programmed death-1), and anti-PD-L1(ligand programmed cell death ligand 1) antibodies have enabled long-term survival in patients for whom the only therapeutic option was previously chemotherapy [5]. The undeniable advantage of immunotherapy is the ability to obtain long-term benefits from treatment. Unfortunately, not all patients receiving immunotherapy obtain benefits from this treatment. PD-L1 expression in tumor or immune cells emerged as the first predictive biomarker of sensitivity to immune checkpoint inhibitors [6]. So far, conducted analyses have shown the possibility of obtaining benefits in PD-L1 negative patients. Approximately 10% of PD-L1-negative cancer patients demonstrate a tumor response, and PD-1/PD-L1 monotherapy could reduce the risk of death in this population by 14% [7]. Chemoimmunotherapy is another option for cancer treatment that combines traditional chemotherapy with immunotherapy. The basis of this connection is the knowledge that chemotherapy can stimulate antitumor immunity and potentiate the clinical activity of immune checkpoint inhibitors (ICIs) [8]. Another group of molecules that have recently attracted interest in cancer therapy are immunoreceptors of T cells with the immunoglobulin domain ITIM (immunoreceptor tyrosine-based inhibitory motif), TIM-3 (T-cell immunoglobulin- and mucin domain-3-containing molecule 3), and LAG-3 (lymphocyte activation gene-3) [9]. The combination of anti-PD-1 and anti-CTLA-4 blockade has proven the possibility of strengthening the anti-tumor response by acting via two separate mechanisms [10]. Adding additional checkpoints to PD-1 blockade offers hope for further improvement in the effects of the treatment of patients and expanding the group responding to immunotherapy.

## 2. Introduction to Immunotherapy NSCLC

In the work of Schreiber R et al., three main mechanisms were indicated by which the immune system plays a protective role in the process of carcinogenesis. The first is the ability to fight viruses, some of which have pro-oncogenic potential. The next is the fight against inflammation, which is a favorable microenvironment for the development of cancer, and the third of the presented mechanisms is the ability to eliminate cancer cells recognized as abnormal, thanks to antigens present on the surfaces of their cells [11]. The last of the presented mechanisms allowed for the creation of breakthrough therapy. The principles of action of checkpoint inhibitors are based on the use of the ability of the immune system to eliminate abnormal cells, including cancer cells, in a process called immune surveillance [9]. Checkpoints are negative regulators of the immune system, which ensure a balance between the elimination of abnormal cells and protection against autoimmune damage to normal tissues. PD-1 is located on the surface of T lymphocytes, B lymphocytes, and NK cells. Physiologically, the combination of the receptor with the PD-L1 ligand leads to the inhibition of T lymphocyte autoreactivity and the conversion of effector T lymphocytes into regulatory lymphocytes (Treg). When the PD-L1 ligand on the surface of cancer cells binds to the PD-1 receptor, the activity of helper lymphocytes (Th) and cytotoxic lymphocytes (CTLs) is inhibited [12,13,14]. Understanding the mechanism of cancer cell escape from the immune system allowed for the creation of specific antibodies that, by blocking PD-1 molecules (pembrolizumab, nivolumab), PD-L1 (atezolizumab, durvalumab), and CTLA-4 (ipilimumab), restore normal activity of the immune system [13,14]. Despite the enormous progress in oncology made by checkpoint inhibitors, not all patients benefit from immunotherapy, which is why efforts are still underway to expand the group of patients who respond to treatment. One of the main problems limiting the efficacy of checkpoint inhibitors includes resistance mechanisms, which may be primary in patients who have never responded to treatment, and acquired, as observed in patients who initially benefited from the use of immune checkpoint inhibitors. Numerous processes responsible for acquired resistance to treatment have been described, such as the compensatory induction of alternative pathways, such as mucin domain protein 3 (TIM-3), lymphocyte activation gene 3 (LAG-3), decrees radio of TILs (tumor-infiltrating lymphocytes) to Tregs and MDSCs, immunoediting with loss of neoantigens, or aberrant WNT/beta-catenin signaling [15,16]. Differences in the efficacy of ICIs also depend on gender—the higher efficacy in male patients and conversely, combination with chemotherapy ICIs is more effective in female patients, provide evidence for the possibility of modifying the response to treatment by strengthening the tumor’s immune environment or enhancing the tumor’s antigenicity [17]. Immunotherapy allows for long-term survival only in a select group of patients [12].

The T cell immunoreceptor with Ig and ITIM domains (TIGIT) also known as WUCAM, Vstm3, and VSIG9 is an inhibitory immune checkpoint receptor that is present on multiple immune cell subtypes, including CD8+ and CD4+ T lymphocytes, NK cells, and Treg cells in both humans and mice [18]. Following the success of anti-CTLA-4, anti-PD-1, and anti-PD-L1 antibodies, TIGIT represents an interesting new therapeutic target alongside lymphocyte activation gene 3 (LAG-3), T cell immunoglobulin, and mucin domain 3 (TIM-3) [19]. Structurally, TIGIT consists of an immunoglobulin variable domain, a transmembrane domain, and an immunoreceptor tyrosine-based inhibitor [20,21]. Studies conducted in vitro and in vivo in mice have demonstrated the possibility of restoring the antitumor response by blocking the TIGIT receptor, leading to increased NK cell cytotoxicity and the activation of CD4+ and CD8+ T cells, as well as the inhibition of Treg activity [22,23,24]. Non-small-cell lung cancer is a subtype of cancer in which TIGIT overexpression has been demonstrated [25]. It may coexist with PD-1 receptors or other inhibitory receptors, such as TIM-3 and LAG-3, on exhausted CD8+ T cells and Tregs in cancer cells [9]. The main ligand for TIGIT is CD155, which belongs to the nectin-like family of molecules and was originally identified as the poliovirus receptor (PVR). CD155 expression is rare in most normal tissues, in contrast to many cancers, where it is overexpressed [26,27]. Physiologically, it plays a regulatory role by binding to the co-stimulatory receptor CD226 and inhibitory checkpoint receptors, such as TIGIT or CD96. In the process of carcinogenesis, it facilitates the growth and migration of cancer cells, and its excessive expression is correlated with tumor progression and poor prognosis [28]. The interaction of TIGIT with CD155 on the surface of dendritic cells inhibits cell maturation, impairing the function and development of T lymphocytes. TIGIT, through binding to CD155, inhibits the CD226 signal, thus promoting cancer cell growth and metastasis by blocking the function of CD8+ T cells and NK cells [29]. Moreover, the study by Leptier A et al. showed lower efficacy of anti-PD-1 therapy in patients with higher CD155 expression in metastatic melanoma [26]. Another low-affinity TIGIT ligand is CD112. The formation of TIGIT association with CD 112 inhibits the proliferation of T cells and NK cells by reducing cytotoxicity and the production of interferon-γ (IFN-γ). Conversely, the formation of the CD112–CD226 complex may promote T cell proliferation and NK cell cytotoxicity, inhibiting the process of carcinogenesis [5,30]. Understanding the importance of TIGIT in the complex regulatory network of positive and negative immunomodulatory receptors on T lymphocytes, tumor cells, and antigen-presenting cells, as well as the negative relationship between CD155 expression and the response to anti-PD-1 treatment, offered hope for the possibility of improving the efficacy of treatment by using further signaling pathways in the immune system [21,29] Figure 1. Lymphocyte activation gene 3 (LAG-3, CD223, FDC) is a class of immune checkpoint receptors that was first described in 1990. The LAG3 gene is located on chromosome 12 (12p13.32). Structurally, it is a transmembrane protein consisting of 498 amino acids, with a transmembrane, extracellular, and cytoplasmic region [31]. The ligands for LAG-3 in the tumor microenvironment are MHC II (major histocompatibility complex class II), galactose-lectin-3 (Galectin-3), hepatic sinusoidal endothelial cell lectin (LSECtin), fibrinogen-like protein 1 (FGL1), and α-synuclein. MHC II is the major ligand for LAG-3, similar to CD4, but with a much higher affinity. Figure 2 shows that the structure of LAG-3 is highly homologous to that of CD4. It is detected in many cell types, including CD4+ and CD8+ T lymphocytes, natural killer T (NKT) cells, natural killer (NK) cells, plasmacytoid dendritic cells (pDCs), and regulatory T cells (Tregs). The upregulation of LAG3 is essential for immune homeostasis and prevention of autoimmunity. Chronic stimulation with tumor antigen leads to the persistent upregulation of LAG3 on tumor antigen-specific CD8+ T cells, which leads to their functional exhaustion, inhibition of T cell proliferation, and cytokine secretion [31,32]. Upon binding of LAG3 to MHC class II, an inhibitory signal is transmitted via its cytoplasmic domain. This association leads to tumor escape from apoptosis and the recruitment of tumor-specific CD4+ T cells, which in turn leads to a decrease in CD8+ T cell responses [33,34]. Another ligand for LAG3 is FGL1, a member of the fibrinogen family. Physiologically, FGL1 is secreted by hepatocytes, but it has been shown that cancer cells can also produce high levels of FGL1. Increased expression of FGL1 mRNA in human solid tumors has been demonstrated in lung cancer (especially adenocarcinoma). Elevated plasma FGL1 levels are associated with poor prognosis and resistance to anti-PD-1 therapy [34]. FGL1 inhibits antigen-specific T cell activation. Cze Wang et al. reported that the blockade of the FGL1–LAG-3 interaction enhances the T cell response and promotes antitumor immunity [35]. Galactin-3 (Gal-3) is a galactose-binding lectin that regulates T cell activation. The formation of LAG3 binding appears to play an inhibitory role in the cytotoxicity of CD8+ T cells. LSECtin may interact with LAG-3 to inhibit tumor-specific T cell responses with the subsequent inhibition of IFNg secretion from antigen-specific effector T cells [36,37,38].

T-cell immunoglobulin and mucin domain-containing-3 (TIM-3) is a type-I transmembrane protein consisting of 281 amino acid residues. The TIM-3 gene is located on chromosome 5 at q33.2. The ligands for TIM-3 are: galactin-9 (Gal-9), high-mobility-group B1 protein (HMGB1), phosphatidylserine (PtdSer), carcinoembryonic antigen cell adhesion molecule 1 (CEACAM-1), and retinoic acid-inducible gene I (RIG-I) TIM-3 is co-expressed on CD4+ and CD8+ lymphocytes with previously described checkpoints, such as PD-1, Lag-3, and TIGIT [9,37,38,39]. In the study by Kaori Sakuishi et al., it was proven that TIM-3-positive tumor-infiltrating lymphocytes with PD-1 co-expression represent the phenotype of the most numerous and most exhausted lymphocytes, unable to proliferate and produce IL-2, TNF, and IFN-γ. Moreover, researchers have shown that the combination of PD-1 and TIM-3 blockade is highly effective in controlling the neoplastic disease and restoring IFN-γ production by T lymphocytes [40]. In the study by Zhuang, X. et al., TIM-3 expression was confirmed on the surface of tumor cells in 86.7% of patients with non-small-cell lung cancer. Importantly, the results of this study suggest that ectopic TIM-3 expression in tumor cells may be a potential, independent prognostic factor [41,42].

## 3. T Cell Immunoreceptor with Ig and Immunoreceptor Tyrosine-Based Inhibitory Motif (ITIM) Domains

### 3.1. Vibostolimab

MK-7684 is a humanized immunoglobulin G (IgG1) monoclonal antibody that, by binding to TIGIT, blocks interactions with its ligands CD112 and CD155, and engages the Fcγ receptor on myeloid cells, leading to the expression of cytokines and chemokines [9,43]. The first study demonstrating its safety and efficacy in monotherapy and in combination with pembrolizumab was published three years ago. The objective response rate was 26% in a study focused on patients with non-small-cell lung cancer, naive to PD-(L)1 inhibitors, treated sequentially with vibostolimab and pembrolizumab. TRAEs were reported in 33 patients (85%). The most frequently reported adverse events were pruritus (38%) and hypoalbuminemia (31%). A proportion of 15% of patients who received combination therapy experienced grade 3–4 TRAEs. Three patients experienced adverse events that led to discontinuation of the treatment. The median progression-free survival (PFS) was 9 months in the subgroup of patients with PD-L1 TPS ≥1% compared with 3 months in the subgroup with PD-L1 TPS <1%. In the population of patients refractory to anti-PD-1/anti-PD-L1 inhibitors, the objective response rate (ORR) was only 3% in those treated with monotherapy and 3% in those treated with a combination of two drugs [43]. The KEYVIBE program was established to assess the safety and efficacy of vibostolimab-based therapies in patients with solid and hematological tumors. It included one phase I study, three phase II studies, and five phase III studies. The aim of the program was to assess the utility of vibostolimab-based therapies in multiple clinical diagnoses and at different stages of cancer. The trials dedicated to patients with NSCLC are KEYVIBE 2 (phase II trial), KEYVIBE 3 (phase III trial), KEYVIBE 6 (phase III trial), and KEYVIBE 7 (phase III trial) [44] (Appendix A).

### 3.2. Tiragolumab

Tiragolumab is a fully human IgG1/kappa anti-TIGIT monoclonal antibody that blocks the formation of the TIGIT-CD155 association. The GO30103 study confirmed its safety and efficacy, and the absence of drug interactions and immunogenicity risk in combination with atezolizumab [45]. In the randomized phase II CITYSKAPE study, chemotherapy-naive patients with relapsed or metastatic PD-L1-positive non-small-cell lung cancer (NSCLC) were randomized to receive tiragolumab (600 mg) plus atezolizumab (1200 mg) or placebo plus atezolizumab intravenously once every 3 weeks. In the primary analysis, the ORR in the combination arm was 31.3% vs. 16.2%. The median progression-free survival (PFS) was 5.4 months in the tiragolumab plus atezolizumab arm vs. 3.6 months in the second arm. In the subgroup with high PD-L1 expression (≥50%), there was a difference in favor of tiragolumab compared with placebo (ORR; 69% vs. 24.1%). The rate of serious adverse events was 3% higher in the combination arm, the most common was increased lipase activity (9% vs. 3%) [46]. The effects of the addition of tiragolumab to atezolizumab and chemotherapy in the SKYSCRAPER-02 trial in patients with small-cell lung cancer were disappointing. The median progression-free survival was 5.4 months in the tiragolumab arm vs. 5.6 months in the control group, and similarly, overall survival was comparable. Treatment-related AEs were reported in 93.3% vs. 92.3% of patients in the tiragolumab and control arms. Grade 3/4 TRAEs occurred in 52.7% of patients in the tiragolumab arm and 55.7% in the control arm. The most common serious TRAEs were anemia and neutropenia [47]. In November 2024, Roche published interim results from the next SKYSCRAPER-01 study, a global, double-blind, phase III study evaluating tiragolumab plus Tecentriq versus Tecentriq alone in patients with PD-L1-high previously untreated, locally advanced, unresectable or metastatic non-small-cell lung cancer (NSCLC). The study did not meet its primary endpoint of overall survival at the final analysis. No new adverse drug reactions of concern were reported during the study. The detailed report of the study will be presented in 2025 [48,49].

### 3.3. Domvanalimab

ARC-7 (NCT04262856) is a randomized, open-label, phase 2 clinical trial that enrolled previously untreated patients with stage IV non-small-cell lung cancer expressing PD-L1 (TPS ≥ 50%). Patients were randomized (1:1:1) to three arms. At a median follow-up of 18.5 months, improvement was demonstrated in the domvanalimab + zimberelimab (anti-PD-1 monoclonal antibody) ± etrumadenant (selective dual antagonist of the A2a and A2b receptors designed to prevent adenosine-mediated immunosuppression) vs. zimberelimab monotherapy [50]. The updated data from ARC-7 includes findings from 150 patients with NSCLC without driver mutation. Patients treated with zimberelimab alone had an ORR (overall response rate) of 30%, of whom 2% achieved a complete response (CR). In the domvanalimab + zimberelimab arm, the ORR was 40%, of whom 2% achieved a CR. The highest ORR achieved in the tripled arm was 44%, but without CR. Immune-related adverse events were reported in 48% of patients (zimberelimab arm) vs. 50% (domvanalimab + zimberelimab arm) vs. 66% in the triplet treated arm. Grade ≥3 treatment-emergent adverse events occurred in 64% of the treated patients in the zimberelimab arm, 46% in the domvanalimab + zimberelimab arm, and 60% in the domvanalimab + zimberelimab + etrumadenant arm. The most common TEAEs were nausea, fatigue, constipation, dyspnea, and pneumonia. Pneumonia was the most common serious TEAE of grade 3 or higher, occurring in 12% of patients. The most common immune-related TEAEs were rashes, occurring in 13 of patients. The rates of pneumonitis in the domvanalimab-containing arms vs. treatment with zimberelimab alone were not noticeably higher [51]. The subsequent ARC-10 trial, which was presented in November 2024 during a poster session at the Society for Immunotherapy of Cancer (SITC) annual meeting, demonstrated improved overall survival (OS) for the combination of domvanalimab with zimberelimab in patients with high PD-L1 expression of non-small cell lung cancer. The presented preliminary results of the study are very promising in both improving the effectiveness of treatment and low toxicity. The 12-month OS rates were 68% for domvanalimab with zimberelimab, 57% for zimberelimab, and 50% for chemotherapy alone, respectively. Serious TRAEs were higher for chemotherapy (47.1%) than for DZ (21.1%) or zimbelimab alone (15.0%). Treatment-related adverse events (TRAEs) leading to treatment discontinuation were higher for chemotherapy (23.5% vs. 10.5% for domvanalimab with zimberelimab arm) [52].

Further studies are ongoing on the use of domvanlimab in patients with non-small-cell lung cancer, including the phase III STAR-121 trial (NCT05502237), comparing the use of domvanlimab and zimberelimab (anti-PD-1 monoclonal antibody) in the first-line setting plus chemotherapy versus pembrolizumab plus chemotherapy. The study is scheduled to be completed by the end of 2027 [53]. Domvanlimab is also being studied in consolidation therapy in the PACIFIC-8 study. PACIFIC-8 is a global study to assess the effects of durvalumab + domvanalimab following concurrent chemoradiation in participants with stage III unresectable NSCLC [54]. Another ongoing study is VELOCITY-Lung substudy-03. A phase 2 study of neoadjuvant domvanalimab + zimberelimab + chemotherapy vs. zimberlimab + chemotherapy followed by adjuvant domvanlimab + zimberlimab or zimberlimab alone in patients with resectable stage II-III non-small-cell lung cancer (NSCLC) is being conducted [55]. It is necessary to wait for the results of these studies.

### 3.4. Belrestotug

Belrestotug is another antibody for which anti-cancer potential was demonstrated in 2018 based on in vitro and in vivo studies. GALAXIES Lung-201 was the Phase II, randomized, open-label study of belrestotug plus dostarlimab in patients with previously untreated locally advanced or metastatic PD-L1 high (TPS >/50%) NSCLC. Treatment-related adverse events were noted in 59% of the dostarlimab-treated group vs. 84% of those receiving belrestotug (400 mg per dose) and dostarlimab. The most common TEAEs, which led to the discontinuation of the treatment were skin (6%) and respiratory disorders (6%). The ORR percentage clearly showed the advantage of using a doublet of drugs (37.5% vs. 65.6%) [56]. Recruitment is currently open for a phase III study GALAXIES Lung-301 examining the combination of dostarlimab with belrestotug vs. pembrolizumab and placebo. The study is planned to be completed in 2029 [57].

### 3.5. Lymphocyte Activation Gene 3 (LAG-3)

Relatlimab is a human IgG4 antibody that restores the effector function of exhausted T cells by binding to LAG-3 [58]. It was the first FDA-approved anti-LAG3 monoclonal antibody in 2022 for the treatment of unresectable or metastatic melanoma in combination with the anti-PD1 monoclonal antibody nivolumab. In the updated RELATIVITY-047 trial in December 2024, the objective response rate (ORR) was 43.7% for the combination of nivolumab with relatlimab and 33.7% for nivolumab monotherapy. Benefit was also noted in terms of PFS of 10.2 months vs. 4.6 months and median OS of 51.0 months vs. 34.1 months, respectively [59]. Success in treating melanoma has initiated research into other cancers also with NSCLC. The current study, NCT05785767, is a randomized, double-blind, phase 2/3 study evaluating fianlimab (anti–LAG-3) plus cemiplimab (anti–PD-1) versus cemiplimab alone in the first-line treatment of advanced non-small-cell lung cancer. Inclusion is conditioned by PD-L1 expression in ≥50% of tumor cells. The study is expected to enroll 850 patients, and the estimated completion year is 2032 [60].

### 3.6. TIM-3

In the phase Ia/b study, the efficacy of the anti-TIM-3 monoclonal antibody LY3321367 in patients with non-small-cell lung cancer varied depending on the response to anti-PD-1/anti-PDL-1 therapy; in patients refractory to anti-PD-1/anti-PDL-1 treatment, the objective response rate to monotherapy was 0% (ORR). Meanwhile, in the group of patients responding to anti-PD-1/L1 treatment, the ORR was 7% [61]. Another interesting option seems to be the use of a bispecific antibody against PD-1 (programmed death-1) and TIM-3 (T-cell immunoglobulin and mucin domain 3) called lomvastomig. This substance simultaneously blocks two different checkpoints. Currently, further studies are underway to evaluate the use of anti-TIM-3 combination with anti-PD1 inhibitors or chemotherapy in solid and in hematological cancers [9].

## 4. Discussion

Despite numerous therapeutic options, the treatment of lung cancer poses a serious challenge to healthcare systems worldwide. The overall survival of patients diagnosed at the metastatic stage remains unsatisfactory. Although chemotherapy is not free from adverse effects and allows for relatively short progression-free periods, it remains the only therapeutic option for many patients [4]. The KEYNOTE-001 study demonstrated the possibility of achieving 5-year survival in patients with advanced non-small-cell lung cancer, with PD-L1 expression on half of the cells in one-quarter of the treated patients [62]. The KEYNOTE-024 study analyzing the efficacy of immunotherapy in the first line of treatment for advanced non-small-cell lung cancer with PD-L1 expression on at least 50% of cells confirmed a significant prolongation of overall survival (OS) compared with the group of patients treated with standard platinum-based chemotherapy (30.0 months vs. 14.2 months) [63]. The CheckMate 227 and CheckMate 9LA studies, which analyzed the treatment with a combination of ipilimumab and nivolumab, were also successful in patients with non-small-cell lung cancer proving that the combination of two mechanisms restoring the antitumor activity of the immune system has an advantage over the use of single immunotherapy [64,65]. Thanks to the results of numerous studies, immunotherapy has become an everyday clinical practice. Lung adenocarcinoma is a cancer with great potential for the use of new checkpoint inhibitors. In the study by Guégan JP et al., a high number of tumor-infiltrating lymphocytes expressing exhaustion markers, such as PD1, LAG3, TIGIT, or TIM3, was shown to be associated with lower objective response rates, shorter progression-free survival, and overall survival. Moreover, their co-expression is associated with resistance to ICIs, regardless of programmed cell death ligand 1 (PD-L1) status [66].

Unfortunately, the NCCN guidelines for the treatment of non-small-cell lung cancer from January 2025 do not indicate the place of drugs using the blockade of the TIGIT, TIM-3, or LAG-3 checkpoints [67]. Despite the great hopes placed on new therapeutic targets, they have limitations. The use of checkpoint blockade is based on the elimination of cancer cells by activated immune cells, which means that tumor-infiltrating lymphocytes (TILs) play a key role in this process. Their effectiveness is limited in patients with reduced immunity and a “cold” tumor microenvironment [68]. Anti-TIGIT antibodies show little activity in monotherapy for advanced solid tumors, despite impressive preclinical results. Studies show that concomitant administration of an anticancer drug with an anti-PD-1 or anti-PD-L1 drug is necessary to achieve antitumor activity [19]. Higher ORR is achieved in patients in the anti-PD-1/PD-L1-naïve setting, which suggests that the optimal time for combination use is before the introduction of ICI therapy in the previous line [69]. A serious problem is the lack of known predictors of the treatment response, and therefore the population in which therapy with drugs blocking two therapeutic targets should be implemented at the cost of increased toxicity is unknown [70]. Clinical data on new ICIs are relatively limited, especially in the case of anti-TIM-3 antibodies, because so far, only a few combinations of anti-TIM-3 antibodies have advanced to Phase III trials, making it difficult to draw definitive conclusions [71].

Currently, numerous studies are underway evaluating the use of new molecules in the treatment of patients in various configurations and clinical indications, including the group of patients with non-small-cell lung cancer [9]. New research directions include the induction of autophagy and apoptosis using cycloastragenol. This study has become the basis for the interest in combination therapies using cycloastragenol and inhibiting the AMPK/ULK1/mTOR pathway in lung cancer cells [72]. The study by Cheng C. et al. provided information on the potential importance of the lower respiratory tract microbiome composition both at the diagnostic stage and in the prediction of progression-free survival (PFS). Further studies are needed to apply this knowledge to clinical practice [73]. The synergistic effect of different treatment strategies that create a favorable environment within the tumor and enable the escalation of the immune system against cancer cells has become the basis of further research.

## Figures and Tables

**Figure 1 ijms-26-04096-f001:**
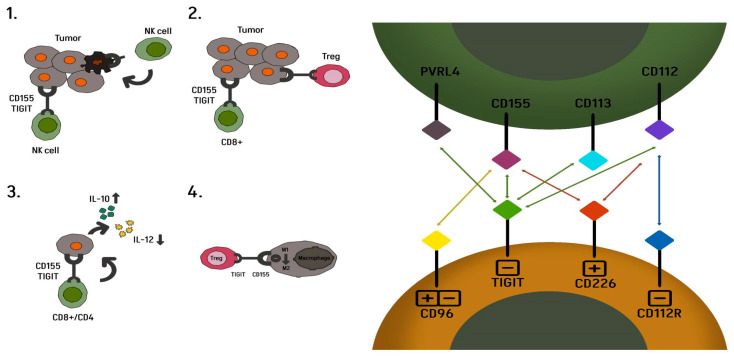
Mechanism of TIGIT action. (**1**) TIGIT inhibits NK cells and promotes their exhaustion. (**2**) Suppression of CD8+ T cell effector function. (**3**) Increasing the production of anti-inflammatory cytokine IL-10 and decreasing the production of pro-inflammatory cytokine IL-12. (**4**) Suppression of CD8+ T cell effector function.

**Figure 2 ijms-26-04096-f002:**
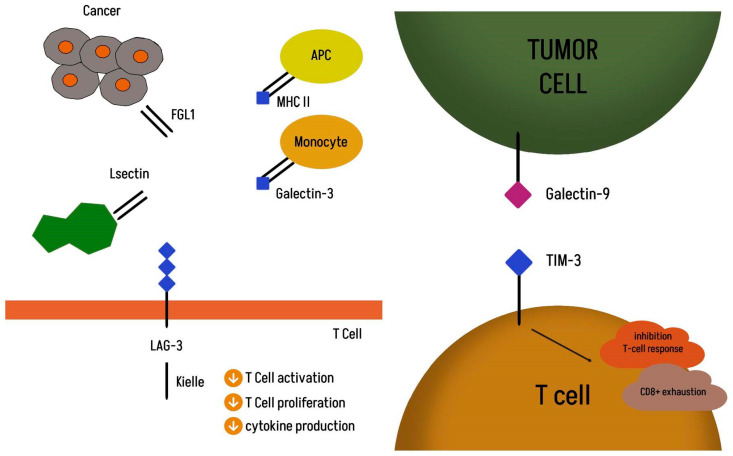
Molecular targets TIM-3 and LAG-3.

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
