# Peer review of "New Therapeutic Targets TIGIT, LAG-3 and TIM-3 in the Treatment of Advanced, Non-Small-Cell Lung Cancer"

_ijms, 2025, doi:10.3390/ijms26094096_

Round 1
Reviewer 1 Report
Comments and Suggestions for Authors
As a major subtype of lung cancer, non-small cell lung cancer (NSCLC) remains a leading health challenge despite the advancements in cancer therapy. Therefore, it would be of critical importance to explore novel immune checkpoints as potential therapeutic targets. In this paper, the authors summarized the recent progress in the studies of targeting TIGIT, LAG-3, and TIM-3 in the treatment of NSCLC. Besides explaining the immunotherapy mechanism, they demonstrated specific clinical trial data and ongoing studies, such as Vibostolimb, Tiragolumab, Domvanalimab et al. even though some of the studies have been discontinued and some clinical data are awaited with hope. The combination of different treatment methods may create synergistic effects, and hold promise for future investigations. This review is relatively comprehensive and a timely summary of the state of the art in this area.
However, before acceptance of the manuscript, the keywords section needs to be reorganized and rewritten. “Coloractal Cancer; Long non-coding RNAs; Gene; Single Nucletide Polymorphism; Genotype” they are irrelevant to the paper.
Author Response
Thank you very much for the review.
We have placed the correct keywords in the attached file.

Reviewer 2 Report
Comments and Suggestions for Authors
In this review paper, the authors listed three potential new targets for cancer immunotherapy treatment, TIGIT, LAG-3 and TIM-3. They collected and concluded some latest researches and clinical traits, emphasizing the possibility for non-small cell lung cancer treatment. Overall, this review paper reported some new development in this field, despite inadequate and outdated. There are some concerns about this article.
Major:
1.The abstract is highly similar to your introduction part, just like cutting from your introduction. The abstract doesn’t clearly claim the aim and significance of this review paper.
2.The keywords selection is not suitable. “Coloractal cancer” is never the focus of this review paper. Other keywords also are not so relevant to this paper.
3.This paper lacks figure and table for results visualization. Some mechanisms are recommended to be illustrated for better presentation. Different clinical traits can also be listed in a table to compare the efficacy.
4.Current results are not adequate and more details should be included in this article. For example, some latest review papers thoroughly explained the developments in this field. Especially, the article “Targeting LAG-3, TIM-3, and TIGIT for cancer immunotherapy” (PIMD:37670328) chose same targets to this paper. The difference between non-small cell lung cancer and other cancers should be emphasized to prove your research novelty, as well as the advantage of targeting these three targets.
5.In Introduction or Discussion, it is recommended to introduce current treatment and research on NSCLC. For example, PMID: 38849220 and PMID: 38884058 could be cited to introduce some relative researches on alternative therapies.
6.Clinical traits results lack rigorous analysis. The significance of existing results and the prospect for the future research and probable problems can be complemented.
Minor:
1.Abstract part, Line 3: “uality” should be “quality”.
2.Introduction part, Line 21: “PDL-1” should be “PD-L1”.
3.Part 2: Introduction to immunotherapy NSCLC part can be subdivided into three parts according to different targets, like “2.1 TIGIT” “2.2 LAG-3” “2.3 TIM-3”.
4.Part 2, last paragraph: Tim-3 should be consistent with previous format “TIM-3” to avoid misunderstanding.
5.Part 3 should first give a brief introduction about the content of this part, and then start to talk about each kind of pharmaceuticals.
Author Response
Thank you very much for the review
1.The abstract is highly similar to your introduction part, just like cutting from your introduction. The abstract doesn’t clearly claim the aim and significance of this review paper.
We have changed the abstract in accordance with your suggestions
2.The keywords selection is not suitable. “Coloractal cancer” is never the focus of this review paper. Other keywords also are not so relevant to this paper.
We have placed the correct keywords in the attached file.
3.This paper lacks figure and table for results visualization. Some mechanisms are recommended to be illustrated for better presentation. Different clinical traits can also be listed in a table to compare the efficacy.
We have included one table summarizing the TIGIT results and 2 figures in the article.
4.Current results are not adequate and more details should be included in this article. For example, some latest review papers thoroughly explained the developments in this field. Especially, the article “Targeting LAG-3, TIM-3, and TIGIT for cancer immunotherapy” (PIMD:37670328) chose same targets to this paper. The difference between non-small cell lung cancer and other cancers should be emphasized to prove your research novelty, as well as the advantage of targeting these three targets.
We have expanded the discussion as recommended.
5.In Introduction or Discussion, it is recommended to introduce current treatment and research on NSCLC. For example, PMID: 38849220 and PMID: 38884058 could be cited to introduce some relative researches on alternative therapies.
We have expanded the discussion as recommended.
6.Clinical traits results lack rigorous analysis. The significance of existing results and the prospect for the future research and probable problems can be complemented.
We have expanded the discussion as recommended.
Minor:
1.Abstract part, Line 3: “uality” should be “quality”.
We have made corrections
2.Introduction part, Line 21: “PDL-1” should be “PD-L1”.
We have made corrections
3.Part 2: Introduction to immunotherapy NSCLC part can be subdivided into three parts according to different targets, like “2.1 TIGIT” “2.2 LAG-3” “2.3 TIM-3”.
In our opinion, the number of subsections in relation to the text would be too large. Therefore, we allowed this part to remain unchanged.
4.Part 2, last paragraph: Tim-3 should be consistent with previous format “TIM-3” to avoid misunderstanding.
We have made corrections
5.Part 3 should first give a brief introduction about the content of this part, and then start to talk about each kind of pharmaceuticals.
In our opinion, this would duplicate the information contained in the introduction and make it more difficult for readers to find answers to the questions they are looking for.

Round 2
Reviewer 2 Report
Comments and Suggestions for Authors
I have no other suggestions.